# Effective Platform Heating for Laser Powder Bed Fusion of an Al-Mn-Sc-Based Alloy

**DOI:** 10.3390/ma16247586

**Published:** 2023-12-10

**Authors:** Dina Bayoumy, Torben Boll, Amal Shaji Karapuzha, Xinhua Wu, Yuman Zhu, Aijun Huang

**Affiliations:** 1Monash Centre for Additive Manufacturing, 15–17 Normanby Rd, Notting Hill, VIC 3168, Australia; dina.bayo@gmail.com (D.B.); amal.shajikarapuzha@monash.edu (A.S.K.); xinhua.wu@monash.edu (X.W.); 2Department of Materials Science and Engineering, Monash University, Clayton, VIC 3800, Australia; 3Karlsruhe Nano Micro Facility (KNMF), Karlsruhe Institute of Technology (KIT), Hermann-von-Helmholtz-Platz 1, D-76344 Eggenstein-Leopoldshafen, Germany; torben.boll@kit.edu

**Keywords:** additive manufacturing, aluminium alloys, LPBF

## Abstract

Platform heating is one of the effective strategies used in laser powder bed fusion (LPBF) to avoid cracking during manufacturing, especially when building relatively large-size components, as it removes significant process-induced residual strains. In this work, we propose a novel and simple method to spare the elaborate post-processing heat treatment typically needed for LPBF Al-Sc alloys without compromising the mechanical properties. We systematically investigated the effects of LPBF platform heating at 200 °C on the residual stress relief, microstructure, and mechanical performance of a high-strength Al-Mn-Sc alloy. The results reveal that LPBF platform heating at 200 °C is sufficient to largely relieve the process-induced residual stresses compared to parts built on an unheated 35 °C platform. Meanwhile, the platform heating triggered the dynamic precipitation of uniformly dispersed (1.5–2 nm) Sc-rich nano-clusters. Their formation in a high number density (1.75 × 10^24^ m^−3^) resulted in a ~20% improvement in tensile yield strength (522 MPa) compared to the build on the unheated platform, without sacrificing the ductility (up to 18%). The improved mechanical properties imply that platform heating at 200 °C can strengthen the LPBF-synthesised Sc-containing Al alloys via in situ aging, which is further justified by an in situ measurement study revealing that the developing temperatures in the LPBF part are within the aging temperature range of Al-Sc alloys. Without any post-LPBF treatments, these mechanical properties have proven better than those of most Al-Sc alloys through long-time post-LPBF heat treatment.

## 1. Introduction 

Over the past few years, laser powder bed fusion (LPBF) has been intensely explored and applied in the additive manufacturing (AM) industry to produce a wide range of alloy systems including aluminium (Al) alloys [1,2]. In comparison with conventional casting, LPBF enables the realisation of complex-shaped components with innovative degrees of freedom and with the highest lead-time efficiency [3]. Compared to other AM technologies, LPBF can achieve a better surface finish and dimensional accuracy [4]. The interest in LPBF-fabricated Al alloys is growing to meet the high demand for critical lightweight structural components for a wide range of applications, such as space, aerospace, and automotive tools and equipment [5,6,7]. 

LPBF enables control of the in-process microstructure of Al alloys through processing parameter optimisation. The rapid cooling rates in LPBF, reaching ~10^6^ K/s (for Al), generate an ultrafine grain structure that supports alloy strength [8]. In addition, the rapidly solidifying tracks result in a highly supersaturated solid solution. A much higher degree of solute supersaturation within the Al matrix can be achieved compared to conventional processing [5]. Subsequently, Al alloys can be solid-solution-strengthened and precipitation-hardened to unprecedented high degrees, which improve the part’s functional performance. For these reasons, many works have reported that the tensile properties of LPBF parts outperform those of conventionally manufactured parts [9,10]. Yet, the fatigue performance of LPBF parts falls short of their conventional counterparts [4]. This has been mainly attributed to the process-induced defects and residual stresses [10,11]. LPBF platform heating has been found to be critical in controlling the microstructure and the mechanical performance [12,13]. The use of a heated platform (160–200 °C) reduced the residual stresses that normally develop in LPBF parts [10,14]. Besides residual stress alleviation, a platform heated to 90 °C has been found to be effective in generating uniform precipitation in the Al matrix [9]. Despite the fact that heating the platform (200 °C) did not improve the mean fatigue strength of Al, it somewhat reduced the fatigue scattering compared to parts built using an unheated platform [10]. The reason for the scattering control has been attributed to the reduction in thermal gradients and cooling rates associated with a preheated platform. As a result, the possibility of stress-induced cracks is lessened.

Producing high-performance Al alloys through LPBF remains challenging, especially for relatively large-scale components. This is because the parts fabricated via LPBF normally contain high levels of thermally induced residual stresses due to the rapid solidification associated with the LPBF process [15]. Moreover, such internal residual stress tends to become exaggerated in large-size components with a prolonged fabrication time. This can lead to crack formation, distortion, delamination, and even the complete failure of the built part [16,17]. To address these issues, using a heated building platform during the LPBF process has been considered one of the effective strategies. For example, previous studies have reported a substantial distortion reduction in the LPBF AlSi10Mg part when the building platform was maintained at 150 °C [18]. Meanwhile, peak hardness of in situ ageing can be achieved at this temperature and, thus, post-heat treatments may not be required. For a higher platform temperature that is more effective for distortion reduction, however, the in situ ageing effect tends to impair the mechanical properties of Al-Si alloys [19,20]. In addition, for the crack-sensitive 2xxx and 7xxx Al alloy series, a heated platform at the temperature range of 100–500 °C has been previously considered [21,22,23,24,25,26]. The high temperature of the platform was applied to lower the thermal gradient and cooling rate during solidification to mitigate the solidification cracks. Despite the density improvement in some cases, producing crack-free parts has still been found to be challenging in these alloys [21,22]. Also, the conventional post-processing T6 treatment did not necessarily improve the LPBF strength compared to their wrought counterparts [27]. Therefore, from these studies, the platform heating strategy for distinct Al alloys is different and requires special considerations in terms of residual stress and distinct microstructural features.

Recently, scandium (Sc)-containing Al alloys specific to LPBF have been developed. These alloys have exhibited an excellent LPBF processability associated with ultra-high strength and, thus, provide great potential over traditional 2xxx and 7xxx Al alloys. Typically, a commercial alloy, known as the Scalmalloy (Al-Mg-Sc-Zr) alloy, has been specifically developed by Airbus Group for AM [8,28]. Owing to the addition of Sc, the unique Scalmalloy can achieve a superior yield strength (YS) reaching beyond 480 MPa after HT [29]. Further studies have shown that LPBF Sc-containing Al-Mn, Al-Mg, and Al-Zn alloys combine superior high strength with YS in the range of 325–625 MPa with an extraordinary level of ductility, reaching ~18% after heat treatment (HT) [5]. Sc-containing Al alloys are designed to develop nano-sized <5 nm) Al_3_Sc-type precipitates that can enhance the strength through a significant age-hardening response in the temperature range of 250–350 °C [28,30,31]. Though this precipitation may potentially occur during LPBF fabrication through the effect of the intrinsic heat treatment experienced by previously solidified material [32,33], it is barely accomplished, considering the ultrafast cooling rates (10^6^ K/s for Al) in LPBF [8]. Therefore, a post-processing heat treatment is commonly needed. An intrinsic heat treatment via platform preheating (200 °C) has been applied to a Sc-modified 5xxx Al-alloy (i.e., the Al-3.4 wt.%Mg-1.08wt.%Sc-0.23wt.%Zr-0.5wt.%Mn-0.44wt.%Cu alloy) [34]. It was found that the strength improved by ~50–100 MPa compared to its counterparts printed using the 35 °C platform. It was further suggested that such a strength improvement was attributed to the coarsening of the Cu-rich particles that are expectedly over-aged at a temperature higher than 250 °C, rather than Al_3_Sc precipitation that typically reaches the peak at similar temperatures [29,34,35,36]. Yet, the nucleation of the Al_3_Sc-type precipitates has been found to commence in the lower temperature range of 200–250 °C [37]. This proposes the feasibility of an in situ ageing effect in Al-Sc alloys during fabrication. 

To preserve the peculiarity of the LPBF Al alloy microstructure, novel heat treatment strategies, other than those conventionally used, need to be explored. The optimum treatment would preserve the microstructure fineness, address the residual stress issues, and improve the mechanical performance. In this study, we explore the possibility of using a platform-heating strategy to enhance mechanical properties and to relieve residual stresses during LPBF fabrication. We first examine the residual stresses that develop in parts printed using 35 °C and 200 °C platforms. Then, the precipitates are characterised and quantified to explain the property enhancements. Through in situ measuring of the temperature of the printed part throughout LPBF fabrication, we show that the attained temperatures are within the aging temperature regime of Al-Sc alloys. Thus, the results of this work are used to recommend a simple in-processing heating method, omitting the time-consuming post-processing artificial aging treatment mostly applied to LPBF Al-Sc alloys, and relieving the high internal residual stresses that typically form within printed parts, while simultaneously maintaining their superior mechanical properties. 

## 2. Methods

### 2.1. Specimen Preparation

Gas-atomised powder was prepared for the alloy composition using a vacuum induction gas atomisation (VIGA) process. The chemical composition was determined via inductively coupled plasma atomic emission spectroscopy (ICP-OES) as Al-2.32Mn-1.42Mg-0.56Sc-0.13Zr-0.03Fe-0.04Si (at.%), i.e., Al-4.58Mn-1.24Mg-0.91Sc-0.42Zr-0.07Fe-0.04Si (wt.%). The powder particle size range was 20–70 µm, with an average size of ~35 µm. The sample processing was performed using a commercial EOS M290 powder-bed machine (Krailling, Germany) equipped with an Yb-fiber laser with a wavelength of 1060–1100 nm, maximum power of 400 W, and spot size of around 100 μm. Samples were built with 350 W (laser power), 1600 mm/s (scan speed), 0.1 mm (hatch distance), and 30 μm (layer thickness), with a laser beam rotation of 67° alternating between consecutive layers. The processing was performed under a controlled argon environment with a minimum oxygen level of 0.1 vol.%. Before the LPBF process was started, the build platform was heated to 35 °C or 200 °C. In both cases, the applied parameters yielded material densification beyond 99.5%. Parts built using the 35 °C platform are designated as AF in this work, while those built using the platform heated to 200 °C are designated as AF-200.

### 2.2. Residual Stress Measurement 

The residual stress measurements were performed at seven points along the build direction on (30 mm × 30 mm × 5 mm) rectangular plates. The sin^2^ψ method of the XRD technique was used to determine the residual stress magnitude in AF and AF-200 plates. The XRD pattern for residual stress measurements was obtained using a Bruker D8 Discover X-ray diffractometer (Billerica, MA, USA), using a Cu-Kα radiation source (λ = 1.5405 Å).

### 2.3. Mechanical Testing

Samples for tensile testing were machined according to ASTM E8/E8M-16a at room temperature using a 100 KN Instron 5982 machine (Norwood, MA, USA) at a strain rate of 0.015 min^−1^. Three horizontal samples (with the tensile axis normal to the build direction) were tested for every condition. The yield strength (YS) is defined as the stress at a plastic strain of 0.2%. 

### 2.4. Microstructure Characterisation 

Samples for scanning electron microscopy (SEM) imaging were cut from the built samples, ground to 2400 Grit size, and polished using silica colloidal suspension. The backscattered electron (BSE) imaging was performed on a JEOL 7001 F field emission gun (FEG) SEM (Tokyo, Japan). For transmission electron microscopy (TEM) imaging, bulk samples were prepared via electro-polishing at −25 °C and 12 volts using a solution of 33% (volume) nitric acid in methanol and then examined in an FEI Tecnai G^2^ T20 Twin LaB 6 TEM microscope (Hillsboro, OR, USA).

The tip samples for atom probe tomography (APT) were prepared using a Zeiss Auriga Dual Beam FIB (Oberkochen, Germany), using Ga+ ions. The region of interest was protected by a layer of Pt before milling. The APT data were acquired under an ultra-high vacuum at a base temperature of 50 K using a Local Electrode Atom Probe (LEAP) Cameca 4000 XR (Gennevilliers, France). The standing high voltage was controlled by the detection rate, set to 0.5%. The device was operated using laser pulsing with a pulse energy of 30 pJ and a 100–200 kHz pulse repetition rate. The APT data were reconstructed and evaluated using Cameca IVAS 3.6.14 software. Isoconcentration surfaces of 2 at.% Sc were used to reveal the precipitates. Second-order nearest neighbour analysis was used to reveal the cluster distribution. 

### 2.5. Powder Temperature Measurement 

Thermocouples were used to measure the temperature at various locations in the part during LPBF processing. For that, a cuboid specimen was built to half height (coloured dark grey in Figure 1). Then, the build was paused, and the powder was removed to install three thermocouples (T1, T2, T3) in embedded (0.9 mm) channels. T1 was located at ~1 mm below the top surface onto which the first layer was deposited (measured from the channel centre to the upper edge of the half-built part). T2 and T3 were separated by ~3 mm along the build direction. Afterward, the powder was backfilled and the LPBF process was resumed. Measurements from thermocouples were collected via a data-logging device every 100 ms. 

## 3. Results

To compare the effect of platform heating for stress relief, the residual stress measurements were first performed across 3 cm profile (as shown in Figure 2) along the building direction for both the AF and AF-200 specimens. The measurement results shown in Figure 2 reveal significant tensile residual stresses, ranging between σ_min_ = 154 ± 4 MPa and σ_max_ = 190 ± 2 MPa, at the different heights from the base plate of the AF material. In comparison, heating the platform to 200 °C reduced the residual stresses of the building process to a lower range of σ_min_ = 59 ± 2 MPa and σ_max_ = 88 ± 3 MPa. 

Given the apparent difference in the residual stress between the AF and AF-200 conditions, we further examined their tensile properties. Due to the fair uniformity of the residual stresses and a lack of severe stress oscillations across the build direction, the tensile properties have been measured at a single height level ~0.8 cm from the platform. The stress–strain curves in Figure 3 reveal an obvious effect of platform heating on the tensile strength. For the AF sample built at the room-temperature base plate, the yield strength (YS) is 438 ± 3.8 MPa, and the total elongation (El) reaches 20 ± 1.2%. In comparison, the specimens built using a preheated platform at 200 °C are strengthened while keeping good ductility. Specifically, the YS reaches 522 ± 3.3 MPa, increased by ~20% compared to the AF sample, while the fracture strain slightly reduces to 18 ± 1%. A yield-drop phenomenon followed by a long plateau is believed to originate from the limited work-hardening capacity associated with the exceptionally refined grains. The underlying mechanism for the evolution of non-uniform plasticity was studied and explained in earlier work [30].

To better understand the origin of the strength improvement of the AF-200 sample compared to the AF counterparts, the microstructures in both conditions were examined and compared. As shown in the SEM-BSE images in Figure 4a,b, the AF-200 sample keeps the typical bimodal grain structure that has been identified in the AF specimen [38]. Alternating regions of fine grains (FGs) and coarse grains (CGs) are observed in the as-fabricated material, whether 200 °C platform heating was applied or not. A close inspection of the grain structure in an FG region of the AF-200 sample reveals that most of the grain sizes are still submicron-sized (<1 μm), similar to the grain sizes in the AF sample that have previously been reported [38].

While there is no apparent difference in the grain structure between the AF and AF-200 samples, we further investigated the secondary phases in the Al matrix. This is because some Sc-rich precipitates, subject to in situ heat treatment, have been suggested in a previous work [39]. Figure 5a provides a BF-TEM image from a CG region of an AF-200 specimen. Although not clear, very tiny bright dots seem to distribute in the matrix. On this basis, weak diffraction spots other than Al have been detected, as marked by the arrows in the corresponding diffraction pattern shown in Figure 5b. Based on the previous work, the position of these diffraction spots indicates the appearance of the L1_2_ structure. 

To confirm the occurrence of tiny precipitates in the AF-200 sample, the APT was used to examine a tip sample taken from a height level of ~0.8 cm from the platform. The AF sample at the same height was also examined for comparison. Isoconcentration surfaces were applied to outline regions of the data comprising an atomic concentration of Sc above 2%. The 3D reconstruction maps for Sc solute (in pink dots), presented in Figure 6a,b, reveal a large volume fraction of fine distribution of Sc-rich precipitates in the AF-200 sample, exhibiting a nearly spherical morphology. Conversely, very limited clusters have been observed in the AF material. 

The proximity histogram in Figure 6c displays the composition profile across the interface between the α-Al matrix and the nano-sized Sc-rich clusters. The profile shows that the clusters consist of Al and Sc and minor other alloying elements. Specifically, the average composition of the Sc-rich clusters comprises 72 at% Al, 20 at% Sc, 3.6 at% Mg, 3.2 at% Mn, and 1.4 at% Zr. This stoichiometry suggests that the clusters have a chemical composition close to the Al_3_Sc precipitates with a relatively high Al/Sc ratio compared to the expected equilibrium stoichiometry of Al_3_Sc. Interestingly, the composition profile confirms that the Sc-rich precipitates are lacking Zr content, i.e., no Zr-rich shell formed onto the Al_3_Sc precipitates.

The existence of the Sc-rich clusters in the AF-200 sample can also be confirmed via nearest neighbour (NN) distribution analyses. The second-order NN for Sc solute atom distance curves from the experimental and random data is provided in Figure 7a,b for AF and AF-200. The NN distance curves revealed a non-random distribution of Sc solute atoms in AF-200, which can be clearly seen in the deviation between the experimental and the random dataset curves. On the other hand, a rather good fit occurred between the experimental and random datasets in AF. The APT analysis reveals the formation of a huge number density (1.75 × 10^24^ m^−3^) of Sc-rich clusters in AF-200 at a volume fraction reaching 0.47%. However, it is worth noting that the cluster size (radius: ~0.86 ± 0.12) in AF-200 is smaller than those observed in the peak-aged alloy (radius: ~1.19 ± 0.3) [30]. In addition, the analysis reveals that ~0.41 at.% Sc is still retained in solid solution in the AF-200 specimen, compared to the nominal composition (0.56 at.%) that has been revealed in the AF sample. In other words, not all Sc atoms have precipitated from the matrix during in situ heating incorporated with platform preheating.

## 4. Discussion

The current work has shown the possibility of further improving the mechanical properties of high-strength Al-Mn-Sc alloys through in-processing treatment. Applying 200 °C platform heating during LPBF fabrication can largely relieve residual stresses and achieve an evident improvement in the tensile strength in comparison with the unheated platform. The residual stress analysis provided in Figure 2 shows that the steep thermal gradients associated with rapid cooling generate large tensile residual stresses, reaching up to 43% of the YS of the AF material. It is worth noting that removing the part from the base platform for residual stress measurement already relieves some of the stress levels. In other words, the actual residual stresses in the parts could be higher than the values displayed in Figure 2. Although elevating the platform temperature to 200 °C did not entirely relieve the residual stresses inherent to LPBF, it reduced the stresses to ~57% compared to the AF material. This is a direct effect of lowering the thermal gradient via platform heating, which reduces the stress levels in the parts. On this basis, heating the platform to 200 °C throughout printing can diminish the crack propagation from the surface during part removal [40,41]. Particularly, large-scale parts that normally experience buckling and distortion due to overheating at long production times can be safely fabricated and removed, given that most of the residual stresses are relieved [19]. 

In addition to residual stress relief, the tensile strength has improved due to platform heating. Compared with the Al-Sc alloys fabricated via LPBF and heat-treated, our AF-200 exhibits an attractive combination of strength and ductility even without any post-processing heat treatment (Figure 8). Its YS exceeds most of the Al-Sc alloys, including the high-strength commercially existing Scalmalloy. Moreover, the ductility surpasses most heat-treated Al-Sc alloys by a significant margin. Interestingly, the AF-200 exhibits comparable properties with a similar alloy system in the peak-aged condition [5]. Achieving such superb tensile properties, in the as-fabricated state, contributes to time and energy savings, because it spares the traditional post-processing artificial aging treatment that is mostly employed for Al-Sc alloys. Since imperceptible change has been observed in the grain structure of AF-200 compared to AF, as shown in Figure 4a,b, probably due to the large number density of thermally stable Sc-rich grain boundary particles stabilising the microstructure against growth [28,42], the secondary precipitation of Sc-rich nanoparticles can be reasonably considered the main origin of the strength improvement in AF-200. During LPBF processing, the underlying solidified material experiences an in situ heat-treatment effect. The effect of such heat treatment depends on the elemental composition of the alloy, in addition to the LPBF processing condition, since the latter dictates the cooling rate. The nucleation of Sc-rich clusters induced via cyclic heating has been observed in the AF specimens prior to platform heating. The APT analysis in Figure 6a reveals the formation of tiny <2 nm Sc-rich precipitates in the solidified AF material. Typically, the ultrafast cooling rate in LPBF (~10^6^ K/S for Al [8]) increases the solute (Sc) supersaturation and, hence, the drive for precipitation upon heating [43]. However, the Sc clusters observed in the AF material indicate that the cooling rate might not be enough to completely trap all Sc in a solid solution. Perhaps, localised regions within each melting pool have been re-melted and, hence, experienced longer heating times, which trigger the decomposition of Sc from supersaturated solid solution. Therefore, the Sc-rich clusters in AF are not as homogenously distributed as in AF-200; compare Figure 6a,b.

These observations suggest that preheating the platform to 200 °C can induce secondary Al_3_Sc precipitation. The formation of Al_3_Sc clusters, verified using the APT data in Figure 6b,c, confirms that the high number density of Al_3_Sc particles (1.75 × 10^24^ m^−3^) has been generated via the effect of the in situ heat treatment using a 200 °C preheated platform compared to AF. The exceptionally high number density of Al_3_Sc hardening particles (Figure 6b) improved the tensile strength compared to AF samples. Being closely spaced and uniformly distributed, these fine clusters effectively resist dislocation motion without sacrificing ductility. 

During the build of the part, the consolidated material heats up because of the high-power laser beam constantly scanning the consolidation powder layer. Most of the absorbed heat from the laser beam will be dissipated downwards through the solidified material to the build platform rather than to the surrounding powder. Therefore, heating the platform to 200 °C reduces the cooling rate, and the heat could be retained in the consolidated material for a longer time. We measured the temperature experienced by the part during the LPBF build by attaching thermocouples at three positions along the build direction. Though T1 is relatively distant (~1 mm, ~33 deposited layers) from the deposition, the time–temperature profile in Figure 9a exhibits an apparent increase in the temperature of the solidified material to a level beyond 200 °C throughout the 100 min print. The enlargement of the first couple of seconds (Figure 9b) shows that the temperature of T1 rises to 270 °C, then settles down to slightly higher than 200 °C. It is reasonable to presume that the top surface layer (~33 layers away from T1) experiences a temperature beyond 270 °C. These rapid thermal cycles are stimulated by the ultrafast laser scan speed (1600 mm/s in this work). 

Knipling et al. [37] have found that the precipitation of Al_3_Sc occurs between 200 and 250 °C in a binary Al-0.1Sc alloy and reaches its peak at 325 °C. Figure 9a shows that for a 30 min scan, the T1 temperature reaches up to 240 °C. This temperature level is enough to induce the dynamic precipitation of Al_3_Sc particles. The repetitive cyclic heating improves the diffusion of the solute atoms. With the existence of a high driving force for precipitation, the nucleation of Sc clusters is triggered. However, the Sc retained in the matrix (up to 73% of the nominal composition), as confirmed via APT analysis, in addition to the small size of the Al_3_Sc particle (radius = 0.86 nm compared to 1.2 nm in the peak-aged material [30]), implies that not all Sc precipitate due to the intrinsic heat treatment. This is mostly the reason behind the relatively lower tensile yield strength (522 MPa) achieved by AF-200 samples compared to peak-aged counterparts (559 MPa) [30]. The amount of Sc retained in the matrix can still precipitate new particles or coarsen the already formed particles in AF-200. Therefore, it is worth exploring the precipitation behaviour at higher energy inputs, such as heating the platform (beyond 200 °C) and reducing the scan speed (below 1600 mm/s).

Notably, the size of the Al_3_Sc precipitates reported here is much smaller than the precipitate size formed through the effect of the intrinsic heat treatment in DED-fabricated Scalmalloy [39]. They reported a radius of 8.1 ± 2.7 nm at the second layer close to the platform and 1.9 ± 0.4 nm at the seventh layer from the platform, while the radius identified here is 0.86 ± 0.12 nm at ~8 mm from the platform. This is mainly attributed to the lower solidification rates in LPBF compared to DED. It can be noted from the temperature profile in Figure 9c that the temperature fluctuations somewhat settle down, to 210–230 °C, in all thermocouples after the first hour. This is due to the highly localised heating source in LPBF compared to DED, in addition to the inherent high thermal conductivity of Al. Both factors improve the rate of heat dissipation in LPBF. This suggests that finer and steadier Al_3_Sc cluster sizes can be achieved during LPBF processing compared to DED. In addition, this temperature level of <230 °C experienced by the underlying consolidated layers (see T3 in Figure 9a) cannot lead to over-aging in the printed Al-Sc parts regardless of the printing time, unlike other Al alloy systems that can reach the maximum strength at 120–200 °C [19]. 

The spherical morphology of the (~1.7 nm) Sc-rich clusters identified in the Al matrix of AF-200, displayed in Figure 6b, is completely different from the cubic morphology of the (~50 nm) intragranular primary and the (~100 nm) grain boundary Sc-rich particles identified in the AF material [42]. Their superfine structure and nearly round morphology imply that these secondary precipitates mainly form through solid-state phase transformation from supersaturated Al matrix during the cyclic heating of the solidified material. The chemical composition of the Sc-rich clusters identified in the AF-200 specimen is not identical to those observed in conventional processing. The ratio of Al to Sc identified here is 3.6, which is higher than the typical ratio of 3 identified in casting [49]. Such an increase in the concentration of Al atoms can be attributed to the ultrafast cooling rate in LPBF processing, which does not allow enough time for Sc atoms to diffuse to Sc-rich clusters and form the equilibrium stoichiometry of Al_3_Sc precipitates. However, care must be taken, as the trajectory aberration during ion evaporation might also create artifacts that affect the APT measurement [50]. The absence of Zr shell around the Sc-rich clusters observed in this work has been similarly observed in peak-aged Al-Mn-Sc alloys [30]. We suggest that this feature may be attributed to (i) the ultrafast cooling rate that does not give sufficient time for Zr to diffuse to Sc-rich clusters and form an enriched atomic shell onto them, considering the slow diffusivity of Zr in Al, (ii) Zr might be entrapped within large intragranular Sc-rich particles, depriving the matrix of Zr atoms [39,42], (iii) higher temperature levels (>325 °C) are needed for the precipitation of Zr-rich shell onto Sc-rich clusters [37]. Though such a temperature level might be attained, it will not be retained for more than a couple of seconds, as can be predicted from Figure 9b. 

## 5. Conclusions

This study presents findings regarding the in situ aging of Al-Sc alloys during LPBF fabrication. Intrinsic heat treatment by means of platform heating at 200 °C proved to largely relieve residual stresses and trigger massive nanoprecipitation during the LPBF fabrication of an Al-Mn-Sc alloy. The following conclusions can be drawn:The residual stresses inherent to LPBF have been measured and compared for parts built on 35 °C and 200 °C platforms. The measurements confirm the presence of high tensile residual stress, reaching up to 40% of the yield stress. These residual stresses have been alleviated and reduced by more than half through platform heating at 200 °C.The evolution of Sc-rich nano-clusters during in situ aging using a 200 °C platform was captured via an atom probe tomography study, from which a high number density (~1.75 × 10^24^ m^−3^) of uniformly dispersed Al_3_Sc nanoparticles (1–2 nm) was estimated.The precipitation and evolution of the nano-scale Al_3_Sc phase formed from the effect of platform heating improved the tensile strength (yielding strength beyond 520 MPa) and excellent ductility (~18%, compared to the 440 MPa achieved using an unheated platform).An in situ measurement study revealed that the temperatures attained within parts built using a 200 °C platform are within the aging temperature regime of Al-Sc alloys. However, the in situ heating effect based on the parameters employed in this work has not been found sufficient to precipitate all the Sc from the matrix and achieve the maximum strength.

## Figures and Tables

**Figure 1 materials-16-07586-f001:**
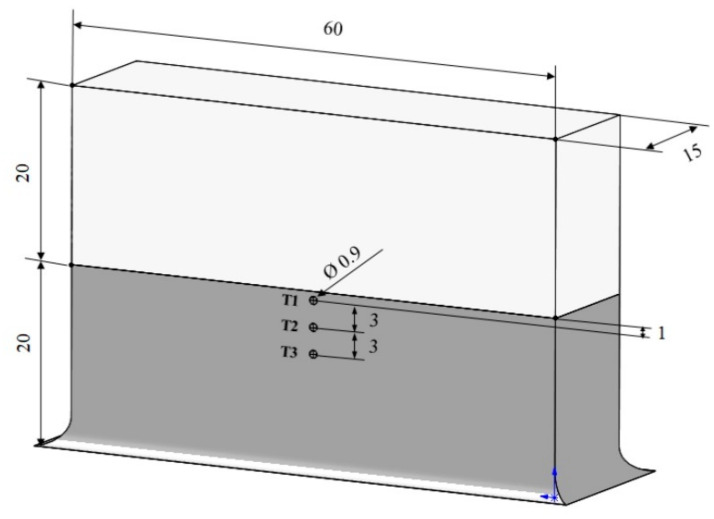
Schematic diagram showing the built specimen with a cuboid shape. There are 3 embedded thermocouple channels shown in the dark grey region (dimensions are in mm).

**Figure 2 materials-16-07586-f002:**
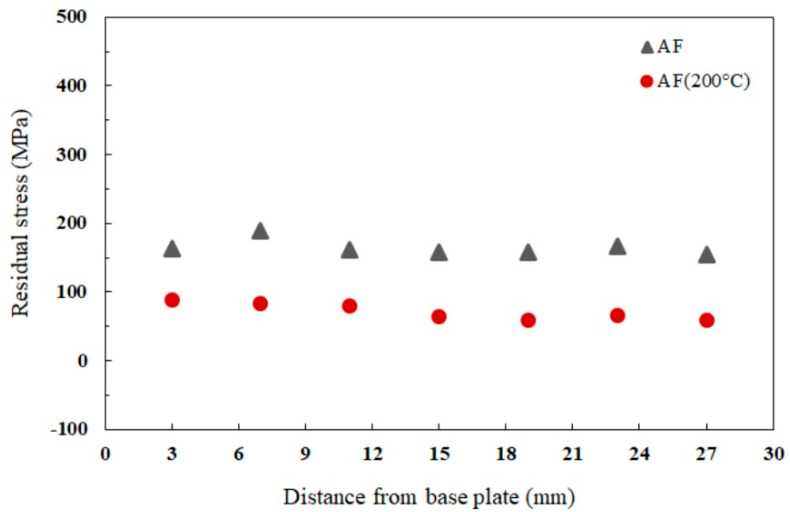
Residual stress distribution within AF and AF-200 specimens measured along the building direction (the standard deviation of the collected data is <4 MPa).

**Figure 3 materials-16-07586-f003:**
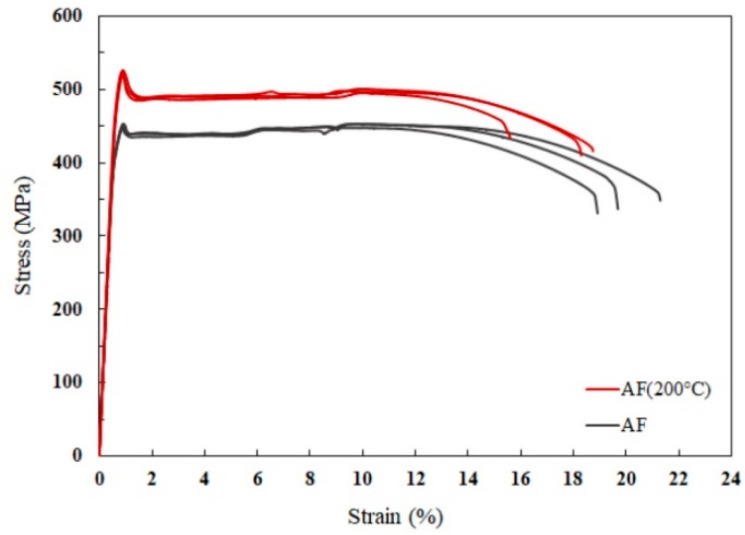
Engineering stress–strain curves of the alloy produced in the AF and AF-200 conditions.

**Figure 4 materials-16-07586-f004:**
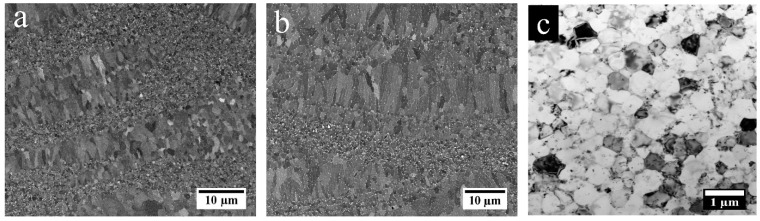
SEM-BSE image showing (**a**) AF and (**b**) AF-200; (**c**) bright-field TEM image for an FG region in AF-200.

**Figure 5 materials-16-07586-f005:**
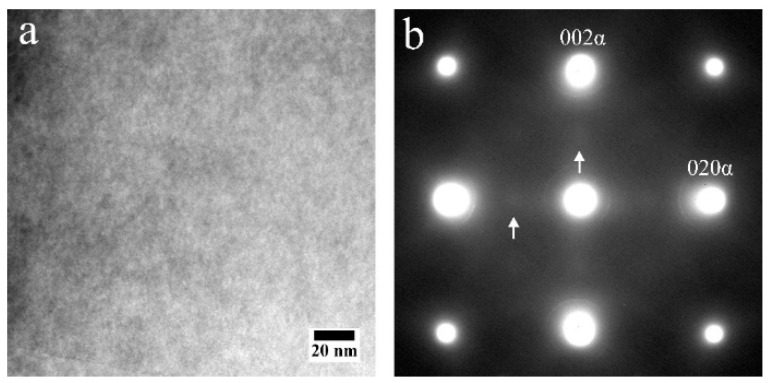
(**a**) Bright-field TEM image from a CG region in an AF-200 sample viewed along the <100>_α_ beam direction; (**b**) the corresponding selected-area electron diffraction pattern from (a), showing weak L1_2_ reflections at {001}_Al_ diffractions, as shown by the arrows.

**Figure 6 materials-16-07586-f006:**
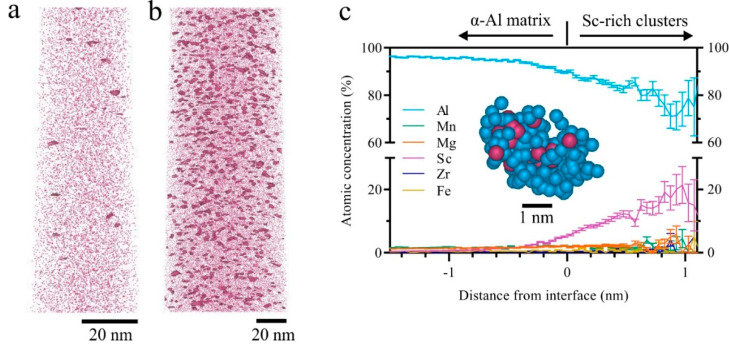
APT reconstruction revealing the solute and precipitate distributions of Sc in (**a**) AF and (**b**) AF-200 tip samples. (**c**) Proximity histogram, combining 1000 precipitate showing the composition profile of Sc-rich precipitates in AF-200 tip. The inset image shows a typical atomic configuration of a Sc-rich precipitate from the AF-200 tip, with Sc atoms shown in pink, and Al in blue.

**Figure 7 materials-16-07586-f007:**
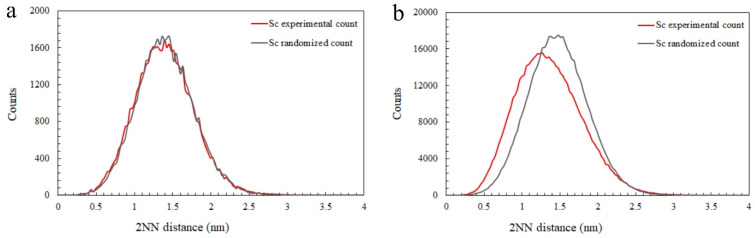
The second-nearest neighbour analysis revealing the deviation between the experimental and the random distribution of Sc atoms in the (**a**) AF and (**b**) AF-200 tip.

**Figure 8 materials-16-07586-f008:**
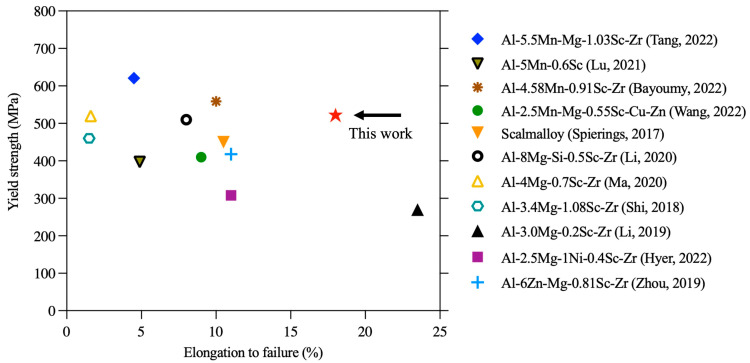
The tensile properties of AF-200 compared with existing Sc-containing Al alloys fabricated via LPBF (all subjected to post-processing heat treatment) [29,30,31,34,35,36,44,45,46,47,48].

**Figure 9 materials-16-07586-f009:**
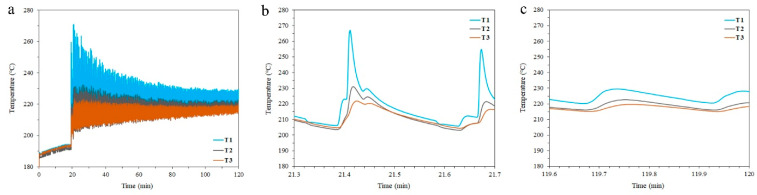
(**a**) Thermocouple measurements during laser passes through the LPBF process, enlarged for clarity: (**b**) 24 seconds from the start of the scan and (**c**) the last 24 seconds in the 100 min print.

## Data Availability

Data are contained within the article.

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
