# Peer review of "Effective Platform Heating for Laser Powder Bed Fusion of an Al-Mn-Sc-Based Alloy"

_materials, 2023, doi:10.3390/ma16247586_

Round 1

Reviewer 1 Report

Comments and Suggestions for Authors

The paper is interesting and covers  a detailed approach taken by the authors.

But the abstract and conclusions could be written in a much better way especially the conclusions. Abstract is cut and short, the key aim of the work i not reflected. Recommended to rewrite the abstract signifying the importance of the study performed in this work. With regards to the conslusions, rather than highlighting only the observations, I would recommend a good summary of the results with key observations.

I would also recommend a graphical abstract showing the key highlights of the work 

Reviewer 2 Report

Comments and Suggestions for Authors

In this work, the influence of platform heating at 200C on the residual stress release, microstructure and mechanical performance in a LPBF Al-Mn-Sc alloy was fully studied. The results are interesting and supported each other. However, there are some few points to be clarified as below:

1.      In this work, only the horizontal samples were tested for the mechanical properties. As well-known, the mechanical property in AM is not uniform in different directions. Therefore, it is suggested to measure the mechanical properties from other directions.

2.      In Fig. 3 of tensile curves, it seems there is a peak before reaching the UTS and even YS. In this case, it is important how to define YS, please explain. Meanwhile, please explain the reason for the peak before 2% of strain.

3.      In Fig. 4, it seems the grain structure between layers in Fig. 4b is different with that in Fig.4a. Please explain the reason.

4.      In Fig. 5, please try other methods to clearly show the presence of Sc-rich precipitates, such as two-beam dark field.

5.      In this work, the platform heating is at 200 C with the not-fully precipitation of Al3Sc PPTs. Then how the authors think about to increase the temperature of platform heating. Could this lead to the fully or more Al3Sc PPTs?

Comments on the Quality of English Language

N/A

Author Response

In this work, the influence of platform heating at 200C on the residual stress release, microstructure and mechanical performance in a LPBF Al-Mn-Sc alloy was fully studied. The results are interesting and supported each other. However, there are some few points to be clarified as below:

The authors thank the reviewer for the positive comment.

  1. In this work, only the horizontal samples were tested for the mechanical properties. As well-known, the mechanical property in AM is not uniform in different directions. Therefore, it is suggested to measure the mechanical properties from other directions.

Anisotropic mechanical properties observed in LPBF aluminium alloys were due to large columnar grains that preferably grow along the building direction. However, the obtained equiaxed-columnar bimodal fine grain structure that form in the Al-Mn-Sc alloys offer great benefits of mitigating the property anisotropy of LPBF fabricated parts. Vertical and horizontal samples have been tested for a similar alloy composition in our earlier work and no significant mechanical anisotropy has been found. Please refer to section 3.3 in “Q. Jia, P. Rometsch, P. Kürnsteiner, Q. Chao, A. Huang, M. Weyland, L. Bourgeois, X. Wu, Selective laser melting of a high strength Al-Mn-Sc alloy: Alloy design and strengthening mechanisms, Acta Mater. 171 (2019) 108–118.

  1. In Fig. 3 of tensile curves, it seems there is a peak before reaching the UTS and even YS. In this case, it is important how to define YS, please explain. Meanwhile, please explain the reason for the peak before 2% of strain.

- The yield strength has been defined in section 2.3. Please see the edited parts highlighted in the manuscript.

-  A concise explanation has been provided in section 3. However, providing a full explanation for the origin of the yield drop occurring in the tensile curve is beyond the scope of this paper. We had devoted an earlier work for explaining this phenomenon, where a thorough explanation for the microstructural features leading to the non-uniform plasticity in the Al-Mn-Sc alloys was provided. Please refer to “D. Bayoumy, K. Kwak, T. Boll, S. Dietrich, D. Schliephake, J. Huang, J. Yi, K. Takashima, X. Wu, Y. Zhu, A. Huang, Origin of non-uniform plasticity in a high-strength Al-Mn-Sc based alloy produced by laser powder bed fusion, J. Mater. Sci. Technol. 103 (2022) 121–133.”

  1. In Fig. 4, it seems the grain structure between layers in Fig. 4b is different with that in Fig.4a. Please explain the reason.

The process-induced microstructure is quite heterogeneous in LPBF Al-Mn-Sc alloy as well as other Al alloys. Different microstructural regions might have different features. Therefore, features might differ as we move from region to another to capture SEM figures. Yet, the comparable feature that can be observed in both Fig. 4 (a) and (b) is the bimodal microstructure consisting of an equiaxed superfine region and a relatively coarse region. This has been observed all over the microstructure.

  1. In Fig. 5, please try other methods to clearly show the presence of Sc-rich precipitates, such as two-beam dark field.

We do use other method, i.e., APT, that has clearly shown the presence of Sc-rich precipitates in Fig. 6(b-c). Due to the tiny size of these Sc-rich precipitates, two-beam dark field cannot show better contrast than the bright field. This is also supported by the pretty weak diffraction spots in the corresponding diffraction pattern in Fig. 5(b), and the reason why we performed the APT to study these precipitates.

  1. In this work, the platform heating is at 200 C with the not-fully precipitation of Al3Sc PPTs. Then how the authors think about to increase the temperature of platform heating. Could this lead to the fully or more Al3Sc PPTs?

More experimental work has to be done in order to explore the possibility of fully aging the Al-Mn-Sc alloy during LPBF processing via controlling the processing parameters. This has been stated in the manuscript in line 354-356 as “Therefore, it is worth exploring the precipitation behaviour at higher energy inputs, such as heating the platform (beyond 200 °C) and reducing the scan speed (below 1600 mm/s).”

Reviewer 3 Report

Comments and Suggestions for Authors

This manuscript presented an interesting work about an effective platform heating for laser powder bed fusion of Al-Mn-Sc based alloy. The work has potential and was correctly carried out. However, some minor points listed below need to be improved.

Introduction: clearer the novelty of this work.

Lines 275-290: I suggest correlate the discussion in this part also with the SEM results presented in Figure 4.

Author Response

Comments of Reviewer 3:

This manuscript presented an interesting work about an effective platform heating for laser powder bed fusion of Al-Mn-Sc based alloy. The work has potential and was correctly carried out. However, some minor points listed below need to be improved.

Introduction: clearer the novelty of this work.

Lines 275-290: I suggest correlate the discussion in this part also with the SEM results presented in Figure 4.

The authors thank the reviewer for the positive comment. A paragraph has been added to the introduction to further elaborate the novelty of this work. In addition, we referred to figure 4 in line 299 as the lines 298-302 discuss the microstructural features observed in figure 4.

Reviewer 4 Report

Comments and Suggestions for Authors

In this paper, the authors conducted research on the effect of LPBF platform heating at 200°C on the residual stress relief, microstructure, and mechanical performance of a high-strength Al-Mn-Sc alloy. They found that platform heating largely relieved the process-induced residual stresses and triggered massive nanoprecipitation during LPBF fabrication, resulting in high strength and excellent ductility. The authors also concluded that further heating is needed to achieve maximum strength.     

However, there are several limitations and shortcomings in the article that should be addressed.

(1) No specific information was provided on the impact of platform heating on other parameters (such as cooling rate, cladding speed, etc.) during the laser powder bed melting manufacturing process.

(2) Although discussed how platform heating affects the mechanical properties and microstructure of materials, did not explore any potential issues or limitations that this heating method may bring.

(3) The impact of platform heating on reducing printing time or improving production efficiency was not mentioned.

Author Response

Comments of Reviewer 4:

In this paper, the authors conducted research on the effect of LPBF platform heating at 200°C on the residual stress relief, microstructure, and mechanical performance of a high-strength Al-Mn-Sc alloy. They found that platform heating largely relieved the process-induced residual stresses and triggered massive nanoprecipitation during LPBF fabrication, resulting in high strength and excellent ductility. The authors also concluded that further heating is needed to achieve maximum strength. However, there are several limitations and shortcomings in the article that should be addressed.

(1) No specific information was provided on the impact of platform heating on other parameters (such as cooling rate, cladding speed, etc.) during the laser powder bed melting manufacturing process.

It is expected that platform heating reduces the cooling rates compared to building at unheated platform. Yet, determining the exact cooling rates is beyond the scope of this work as it is not essential for explaining the observations. Furthermore, platform heating does not necessarily affect the laser scan speed. Optimized processes had been applied in building the parts in this work. The corresponding material densification was found to exceed 99.5% in parts built at 35 °C and 200 °C.

(2) Although discussed how platform heating affects the mechanical properties and microstructure of materials, did not explore any potential issues or limitations that this heating method may bring.

According to the literature, building at a preheated temperature 200 °C might lead to over-aging in some alloy systems such as AlSi10Mg (ageing temperature ~ 160 °C). Please refer to “F. Bosio, H. Shen, Y. Liu, M. Lombardi, P. Rometsch, X. Wu, Y. Zhu, A. Huang, Production Strategy for Manufacturing Large-Scale AlSi10Mg Components by Laser Powder Bed Fusion, JOM. (2021)”. This also depends on the total time of the build. Yet, the in-situ measurement study presented in his work suggest that the attained temperatures are not supposed to lead to over-aging of Al-Mn-Sc alloy system (ageing temperature ~ 300 - 350 °C).

(3) The impact of platform heating on reducing printing time or improving production efficiency was not mentioned.

In this work, similar scan speed 1600 mm/s was found optimum for both 35 °C and 200 °C builds. So, platform heating did not reduce the printing time. Yet, it could improve the production efficiency as it spares the time-consuming post processing heat treatment. The results show that platform heating can largely relieve the residual stresses and simultaneously enhance the mechanical properties.

Round 2

Reviewer 2 Report

Comments and Suggestions for Authors

The comments are well explained with proper modifcations in the text. Then it is suggested to be accepted. 

Reviewer 4 Report

Comments and Suggestions for Authors

The author has carefully revised the manuscript, and the current version is acceptable.